# Antibacterial Efficacy of *N*-(4-methylpyridin-2-yl) Thiophene-2-Carboxamide Analogues against Extended-Spectrum-β-Lactamase Producing Clinical Strain of *Escherichia coli* ST 131

**DOI:** 10.3390/molecules28073118

**Published:** 2023-03-31

**Authors:** Gulraiz Ahmad, Aqsa Khalid, Muhammad Usman Qamar, Nasir Rasool, Malik Saadullah, Muhammad Bilal, Majed A. Bajaber, Ahmad J. Obaidullah, Hadil Faris Alotaibi, Jawaher M. Alotaibi

**Affiliations:** 1Department of Chemistry, Government College University Faisalabad, Punjab 38000, Pakistan; gulchemist35@gmail.com; 2School of Interdisciplinary Engineering & Science (SINES), National University of Sciences and Technology (NUST), Islamabad 44000, Pakistan; aqsa.khalid30@yahoo.com; 3Institute of Microbiology, Faculty of Life Sciences, Government College University Faisalabad, Punjab 38000, Pakistan; musmanqamar@gcuf.edu.pk; 4Department of Pharmaceutical Chemistry, Government College University Faisalabad, Punjab 38000, Pakistan; maliksaadullah@gcuf.edu.pk; 5School of Chemistry and Chemical Engineering, Shandong University, Jinan 250100, China; muhammadbilalgcuf@gmail.com; 6Chemistry Department, Faculty of Science, King Khalid University, Abha 61413, Saudi Arabia; mb@kku.edu.sa; 7Department of Pharmaceutical Chemistry, College of Pharmacy, King Saud University, Riyadh 11451, Saudi Arabia; aobaidullah@ksu.edu.sa (A.J.O.); 441203066@student.ksu.edu.sa (J.M.A.); 8Department of Pharmaceutical Sciences, College of Pharmacy, Princess Nourah bint Abdulrahman University, Riyadh 11671, Saudi Arabia; hfalotaibi@pnu.edu.sa

**Keywords:** ESBL-producing *E. coli*, ST131, blaCTX-M, carboxamides, docking analysis

## Abstract

Development in the fields of natural-product-derived and synthetic small molecules is in stark contrast to the ongoing demand for novel antimicrobials to treat life-threatening infections caused by extended-spectrum β-lactamase producing *Escherichia coli* (ESBL *E. coli*). Therefore, there is an interest in the antibacterial activities of synthesized *N*-(4-methylpyridin-2-yl) thiophene-2-carboxamides (**4a**–**h**) against ESBL-producing *E. coli* ST131 strains. A blood sample was obtained from a suspected septicemia patient and processed in the Bactec Alert system. The isolate’s identification and antibacterial profile were determined using the VITEK 2^®^ compact system. Multi-locus sequence typing of *E. coli* was conducted by identifying housekeeping genes, while ESBL phenotype detection was performed according to CLSI guidelines. Additionally, PCR was carried out to detect the blaCTX-M gene molecularly. Moreover, molecular docking studies of synthesized compounds (**4a**–**h**) demonstrated the binding pocket residues involved in the active site of the β-lactamase receptor of *E. coli.* The result confirmed the detection of *E. coli* ST131 from septicemia patients. The isolates were identified as ESBL producers carrying the blaCTX-M gene, which provided resistance against cephalosporins and beta-lactam inhibitors but sensitivity to carbapenems. Among the compounds tested, **4a** and **4c** exhibited high activity and demonstrated the best fit and interactions with the binding pocket of the β-lactamase enzyme. Interestingly, the maximum of the docking confirmations binds at a similar pocket region, further strengthening the importance of binding residues. Hence, the in vitro and molecular docking studies reflect the promising antibacterial effects of **4a** and **4c** compounds.

## 1. Introduction

Antibiotic-resistant (AMR) infections are growing in number and severity around the world, particularly in developing nations such as Pakistan. If we do not address AMR today, one person will die every three seconds by 2050 [1]. Globally, AMR pathogens caused 1.27 million deaths in 2019, leading to prolonged hospital stays and fatal consequences [2]. Alarming projections suggest that the death toll caused by drug-resistant infections in the upcoming years and decades may surpass the current global fatality rate of the COVID-19 pandemic, which has already prompted multibillion-dollar investments in vaccine development, drug repurposing, and anti-viral discovery [3]. Extended-spectrum β-lactamase (ESBL)-producing infections are listed as “High-Priority Pathogens” by the World Health Organization [4], and the treatment of infections brought on by *Escherichia coli* that produce ESBL (ESBL *E. coli*) is a significant public health concern. Few therapeutic options are available since these organisms resist widely used medicines such as penicillins and cephalosporins [5]. The plasmid-mediated antimicrobial resistance genes blaCTX-M, blaTEM, and blaTEM gene were carried by ESBL-producing *E. coli*. These are in charge of causing severe clinical infections, such as urinary tract infections, bacteremia, septicemia, wound infections, skin infections, and soft tissue infections [6].

The most prevalent lineage of extraintestinal pathogenic *E. coli* (ExPEC) worldwide is ST131. According to historical studies, this lineage is believed to have gained prominence as far back as 2003. In contrast to other classic-group B2 ExPEC isolates, ST131 isolates are often found to produce extended-spectrum β-lactamases, such as CTX-M-15, and have near-universal resistance to fluoroquinolones [7,8,9]. This was initially detected in 2008 [10,11]. Patients with urinary tract infections are most commonly diagnosed with ST131 [12].

Since no novel therapeutic techniques have been developed in the past two decades to treat life-threatening infections, they are crucial in the fight against AMR. Pyridyl is a crucial moiety that exhibits insecticidal, anti-inflammatory, anti-cancer, anti-p38 MAP kinase, anti-viral, and anti-convulsant properties [13,14,15,16,17,18,19]. Nicotinamide adenine dinucleotide (NAD) undergoes redox reactions in biological systems, causing its pyridine ring to be reduced to dihydropyridine and producing NADH. Comparable redox reactions are also found in anabolic processes involving NAD phosphate interconversion (NADP+/NADPH) [20]. Carboxamide is a crucial scaffold with antibacterial properties. Proteins’ crucial building block, the carboxamide bond (-CO-NH-), has drawn much interest because it resists hydrolysis. Because of the ability to create peptides from extremely basic amino acid substrates, this phenomenon is essential in biological systems [21,22,23,24]. The expression of efflux systems that prevent the antibacterial drug from reaching its intracellular target, the modification of the drug’s target site, or the production of an alternative metabolic pathway that avoids the drug’s action are all possible effects of acquired resistance genes in bacteria [24]. ESBLs are enzymes found in Gram-negative bacteria that hydrolyze antibiotics and lead to resistance against third-generation cephalosporins [25]. The main RND efflux complex in *E. coli* is made up of the pump protein AcrB, the outer membrane-crossing channel TolC, and the protein AcrA, which links TolC and AcrB. A proton-motive force drives the efflux pump complex. Due to the significance of the antibacterial properties of pyridyl and carboxamide, which we have previously explored in our research against clinically derived resistant bacteria [26,27,28]. 

In previous discussions, we covered the synthesis of 5-aryl-*N*-(4-methylpyridin-2-yl)thiophene-2-carboxamides (**4a**–**h**), their in vitro cholinesterase inhibition, and their in vivo anti-convulsant activity against Alzheimer’s disease [29]. We expanded our research on *N*-(4-methylpyridin-2-yl)thiophene-2-carboxamides (**4a**–**h**) for their ability to dock with *E. coli* 6N6K and 7BDS proteins in ESBL-producing *E. coli*.

## 2. Results and Discussion

### 2.1. Chemistry

Our research group has previously reported the synthesis of 5-Bromo-*N*-(4-methylpyridin-2-yl)thiophene-2-carboxamide (**3**) in good yield (80%) by reacting 5-bromothiophene-2-carboxylic acid (**1**) with 2-amino-4-methylpyridine (**2**) in the presence of titanium tetrachloride (TiCl_4_) coupling reagent and pyridine which acts as solvent as well as a base. A library of 5-Bromo-*N*-(4-methylpyridin-2-yl)thiophene-2-carboxamides (**4a**–**h**) has been synthesized in moderate to good yields of 35–84% by the Suzuki Miyaura cross-coupling reaction of the intermediate compound (**3**) with various aryl boronic acids, using commercially available Palladium tetrakis catalyst (Figure 1, Figure 1) [29].

### 2.2. Confirmation of the Bacterial Isolate

The clinical isolate was confirmed as *E. coli* from the blood sample and belonged to the pathogenic strain ST131. *E. coli* ST131 was phenotypically positive for ESBL production and contained the blaCTX-M gene.

### 2.3. Antibacterial Activity

ESBL-producing *E. coli* ST131 displayed resistance to the AWaRe (Access, Watch, and Reserve) WHO classes of antibiotics. They showed increased MIC (µg/mL) to ampicillin (≥512), ceftazidime (≥256), ceftaroline (≥8), ciprofloxacin (≥8), and cotrimoxazole (≥4/152). However, the most sensitive drugs were imipenem (2), nitrofurantoin (16), and fosfomycin (32), as presented in Table 1. The isolate obtained from the patient blood sample was confirmed as *E. coli* by VITEK 2^®^ compact system. The isolate was synergistically resistant to cephalosporins antibiotics and monobactam and sensitive to carbapenems and polymixins.

The molecules (**4a**–**h**) were screened for antibacterial activity at five concentrations (10, 20, 30, 40, and 50 mg/well) against ESBL-producing *E. coli* by agar well diffusion. Broth dilution methods calculated MIC and MBC. The results are presented in Table 2 and Figure 2 and Figure 3 regarding the activity. The zone of inhibition (mm) increases as the concentration of compounds increases. Compounds **4a** and **4c** showed the highest zone of inhibition (13 ± 2 and 15 ± 2, respectively) at 50 mg concentration of compound as compared to other compounds. All other compounds also showed good activity. The MIC and MBC of the tested molecules were calculated (Table 3 and Figure 4 and Figure 5).

### 2.4. Molecular Docking Studies

Molecular docking of β-lactamase from *Escherichia coli* strain with carboxamide derivatives, Sakai (PDB ID: 6N9K), demonstrated necessary hydrogen bonding and hydrophobic interactions. In addition, the binding pattern of 6N9K revealed two binding cavities (Figure 6A). However, the maximum of the poses resides in the binding pocket A. Therefore, the binding cavity A confirmations were selected for protein-interaction analysis. Moreover, the superposition of the molecule **4a** confirmations was not so good due to the flexibility and conformational changes in the ligand and protein environment. We have observed that the superposition of different ligand conformations in pocket A can result in variations in the binding pose and binding affinity, leading to inconsistencies in the molecular docking results. Therefore, to address this issue, we performed multiple docking runs using different conformations of the ligands and analyzed the resulting poses and scores. This approach allowed us to obtain a more comprehensive view of the ligands’ potential binding modes and identify the most likely binding poses. In addition, the two highly active compounds **4a** and **4c** showed almost a similar binding pattern, further validating the authenticity and stability of the interacting residues and the binding cavity. The highly active compound **4a** showed hydrophobic interactions with N255, D258 and hydrogen bonding with R257 amino acid residues (Figure 7A). Another compound **4c** in the carboxamide derivates demonstrated hydrogen bonding interaction between the carboxyl (C=O) group and L358 amino acid residue. Moreover, two hydrophobic interactions were also observed between the (i) benzene (6-ring) and I64 residue (ii) thiopyran and P42 (Figure 6B). The interaction pattern of the remaining compounds with 6N9K is shown in Appendix A. 

In the case of the CTX-M-15 enzyme (PDB ID: 7BDS), all the conformations bindings at the similar region (Figure 6B) of the receptor are considered an important site to target for the inhibition of the CTX-M-15 enzyme to control antibiotic resistance. The highly active compound **4a** showed hydrophobic interactions, i.e., thiopyran, chlorobenzene with T105, and pyridine with A219 and R274 amino acid residues. Interestingly, hydrogen bonding interaction was also observed between the carboxyl (C=O) group and S130 amino acid residue (Figure 7C). For compound **4c**, only one hydrophobic interaction was observed between thiopyran and S237. Moreover, two hydrogen bonding interactions also demonstrated (i) OH and S130 (ii) NH and P167 (Figure 7D). Moreover, the interaction pattern of the rest of the carboxamide derivatives with 7BDS is shown in Appendix A. 

Additionally, we compared the docking results of *N*-(4-methylpyridin-2-yl) thiophene-2-carboxamide analogues with tazobactam, the natural ligand of the target protein (7BDS). Our docking analyses showed that tazobactam exhibited hydrogen bonding interactions with R44, E64, R65, A172, and T264 residues, while no hydrophobic interactions were observed (Appendix A). Moreover, the binding energy and RMSD values for tazobactam were found to be relatively higher than those of the synthesized carboxamide analogues (Appendix A). The observed differences in the interaction pattern between tazobactam and the carboxamide analogues may be attributed to their structural diversity. Tazobactam is a β-lactamase inhibitor with a β-lactam ring, while the carboxamide analogues synthesized in this study have a thiophene ring and a carboxamide group. These structural differences may lead to variations in the binding orientations and interactions with the target protein.

Moreover, the carboxamide analogues have additional interactions, such as hydrophobic interactions, due to the presence of the thiophene ring, which may contribute to the stability of the protein-ligand complex. This may explain the relatively lower binding energy and RMSD values observed for the carboxamide analogues compared to tazobactam. Based on these comparisons, we concluded that the synthesized carboxamide analogues exhibited a more stable binding conformation and better interaction patterns than the natural ligand tazobactam.

In conclusion, hydrophobic and hydrogen bonding interactions and the binding pockets for 6N9K and 7BDS identified from this study are important residues and sites that could be used to check the inhibition potential of the designed chemical compounds. Moreover, compounds **4c** and **4d** showed better binding patterns and least energy values (most stabilizing energy conformation) and Root-Mean-Square Deviations (RMSDs) from the original conformation (Appendix A) as compared to the rest of the compound series which is in agreement with the experimental results. These findings suggest that the synthesized compounds have the potential as effective inhibitors for the target protein. Overall, our study provides valuable insights into the binding mechanisms of the carboxamide analogues and paves the way for further optimization and development of novel inhibitors against this protein.

### 2.5. Molecular Dynamic (MD) Simulations of Highly Active Compounds

In this study, we conducted molecular docking and 100 ns molecular dynamics (MD) simulations of the β-lactamase (PDB ID: 6N9K) and antimicrobial protein from Klebsiella (PDB ID: 7BDS) with highly active compounds (**4a**,**4c**) using the CHARMM36 force field in GROMACS. The 6N9K-**4a** docked complex remained stable throughout the simulation, while minor fluctuations in the root-mean-square deviation (RMSD) plot were observed for the 6N9K-4c complex after 80 ns. These minor fluctuations may be attributed to the natural flexibility and dynamics of the 6N9K-**4c** complex. The overall RMSD value of both docked complexes (6N9K-**4a** and 6N9K-**4c**) was less than 0.3 Å, indicating stability and maintenance of their initial conformation throughout the 100 ns simulation (Figure 8A). 

Similar to the 6N9K-**4a** complex, no fluctuations in the RMSD plot were observed for the 7BDS-**4a** complex. However, a significant rise in the RMSD trajectory was observed for the 7BDS-**4c** complex after 80 ns of simulation. This rise in RMSD may be attributed to conformational changes or the instability of the protein-ligand complex. Nevertheless, it is noteworthy that the overall RMSD values for both 7BDS-**4a** and 7BDS-**4c** complexes were less than 0.4 Å, indicating that these complexes were also stable and maintained their initial conformation throughout the simulation (Figure 8B). Our results suggest that both docked complexes (6N9K-**4a** and 6N9K-**4c**) are stable, while the stability of the 7BDS-**4c** complex may require further investigation. These findings may aid in designing novel inhibitors for β-lactamase enzymes, which are critical for combating antibiotic resistance in bacteria.

In addition, our experimental studies showed high IC50 values at 50 mg/mL and inhibition zones of 13 ± 2 for **4a** and 15 ± 2 for **4c**, which agrees with our computational findings. This agreement between computational and experimental results adds strength to our findings. It suggests that our computational approach can be reliable for identifying and evaluating potential inhibitors for β-lactamase enzyme inhibitors. Our study provides a promising approach for identifying potential inhibitors for the β-lactamase enzyme. In the future, we plan to investigate the binding mechanisms of these highly active compounds using more advanced computational techniques and to conduct in vivo studies to validate their effectiveness as antibacterial agents.

## 3. Materials and Methods

### 3.1. Antibacterial Activity

#### 3.1.1. Identification of the Bacterial Strains

After obtaining ethical permission from the Ethical Review Committee (ERB), Government College University Faisalabad, and written consent from the patients, 3 mL blood samples were taken and inserted into the BACTEC/Alert blood culture bottle. The bottles were incubated at 37 °C in BACTEC/Altert automatic system (BD, UK). The positive samples were sub-cultured on blood and MacConkey agar (Oxoid, UK). Using a Gram-negative card, the isolates were preliminarily identified based on colony morphology and biochemically confirmed in an automatic VITEK 2^®^ compact system (Biomerieux, Marcy-l’Étoile, France). 

#### 3.1.2. Antibiogram of the Isolates 

MIC (µg/mL) of various antibiotics against these pathogens was carried out in VITEK 2^®^ compact system (BioMerieux, Marcy-l’ÉtoileFrance France). The antimicrobials tested were ampicillin, pipracillin, amoxicillin/clavulanic acid, ceftazidime, ceftriaxone, cefepime, ceftaroline, aztreonam, imipenem, meropenem, amikacin, ciprofloxacin, levofloxacin, tetracycline, co-trimoxazol, nitrofurantoin, fosfomycin, and colistin (Table 1). The interpretation of the susceptibility was made as per CLSI guidelines 2020. 

#### 3.1.3. Phenotypic Detection ESBL Enzyme

ESBL production was determined by the double disc diffusion method [30]. In short, the isolates were owned on the MHA plate, co-amoxiclav antibiotic was placed in the center, and 3rd generation cephalosporins were placed at a distance of 10 mm to co-amoxiclav. After overnight incubation, the synergy-like pattern confirmed the presence of ESBL enzymes (Figure 9). 

#### 3.1.4. MLST of *E. coli* Strain

For the MLST, seven housekeeping genes (adk, fumC, gyrB, icd, mdh, purA, and recA) were amplified and sequenced according to the conditions provided by the EnteroBase Database. The obtained PCR products were sent for sequencing to Macrogen (South Korea). The ChromasPro software (Technelysium, South Brisbane, Australia) edited the raw sequences. The sequences were aligned using the ClustalW algorithm (MEGA, version number.7 software). The allele number was assigned to each gene locus, and the sequence types (STs) were determined according to each isolate’s allelic profile [31]. 

#### 3.1.5. Molecular Detection of blaCTX-M Gene

Bacterial DNA was extracted by a commercially available DNA extraction kit (Qiagen, Germany). The following primer detected the blaCTX-M gene; blaCTX-M F: ATGTGCAGYACCAGTAARGTKATGGC, blaCTX-R: TGGGTRAARTARGTSACCAGAAYCAGCGG by PCR. The PCR reactions (30 μL) were used using 15 μL 2X DreamTaq Green master mix (Thermo Fisher Scientific, Waltham, MA, USA), 1 μL of each primer (forward and reverse) was used and 1 μL of the template DNA was used. The PCR water was used to make a total volume of 30 μL. The agarose gel (1.5%) was used to visualize the amplicons after electrophoresis at 100 V for 30–45 min. All the amplified samples were sent for Sanger sequencing from Macrogen, Inc. (Gangnam-gu, South Korea). The obtained sequences were aligned and compared with the sequences in the GenBank database using the NCBI BLAST tool. 

#### 3.1.6. Agar Well Diffusion Method of Compounds (**4a**–**h**) against ESBL-Producing *E. coli*

The antibacterial activity of the compounds was determined by agar well diffusion assay against ESBL-producing *E. coli* described previously [28]. In short, 0.5McFarland bacterial suspension was inoculated on the Mueller Hinton Agar plate, and a sterile 6 mm cork borer was used to make wells on each plate. Subsequently, 100 μL of each DMSO diluted compound (50 mg, 40 mg, 30 mg, 20 mg, 10 mg) was poured into each well, and plates were incubated at 37 °C overnight. Vernier caliper measured the zone of inhibition (mm). The assay was performed in triplicate (Table 2, Figure 2 and Figure 3). 

#### 3.1.7. Minimum Inhibitory Concertation of Compounds (**4a**–**h**) against ESBL-Producing *E. coli*


Microbroth dilution assay determined different compounds’ MIC (% *w*/*v*) [28]. Two to three isolated colonies were mixed in 20mL of double-strength lysogeny broth (LB) medium in 50 mL of falcon tubes and incubated at 37 °C overnight. The bacterial suspension was diluted to achieve 0.5 McFarland at an optical density (OD) of 0.07 at 600 nm. Briefly, serial dilutions of each compound (0.76, 1.56, 3.12, 6.25, 12.5, 25, 50 mg) were prepared in DMSO, and 100 μL of each compound dilution was added in 96 wells, flat-bottom microtiter plates (Thermo Fisher Scientific, Leicestershire, UK). Subsequently, 100 μL of bacterial suspension was added to each well. Negative-control wells contained 100 μL of LB, and positive-control wells contained LB with bacterial suspension. The microtiter plate was incubated at 37 °C overnight in a shaking incubator (MaxQTM Mini 4450, Thermo Fisher Scientific) at 3 g. MIC was calculated by comparing each well with negative and positive-control wells. All procedures were performed in triplicate (Table 3, Figure 4 and Figure 5). 

#### 3.1.8. Minimum Bactericidal Concentration of Compounds (**4a**–**h**) against ESBL-Producing *E. coli*


Minimum bactericidal concentration (MBC, % *w*/*v*) is the first dilution with no growth on the agar plate. A 10 μL sample was taken from no-visible-growth wells of a microtiter plate. It was inoculated on the nutrient agar plates (Oxoid, Hampshire, UK) and incubated at 37 °C for 24 h aerobically. Plates were examined for cell viability, and any colonies developed were scored as bacterial growth and no bacterial growth. All procedures were repeated in triplicates (Table 3, Figure 4). 

### 3.2. Molecular Docking 

Hyperproduction of β-lactam antimicrobial agents in *E. coli* inactivate and, thus, provides resistance to the antibiotic used to treat *E. coli* infections [32]. Additionally, *E. coli* strains carrying CTX-M-15 ESBLs are notably observed to be highly prevalent in the Asian population and found to be the most abundant enzyme for causing infectious diseases. Interestingly, previous studies showed that CTX-M-15 also exhibits resistance against third-generation cephalosporin drugs [33,34]. Therefore, in our study, molecular docking studies of synthesized carboxamide derivatives against β-lactamase and CTX-M-15 in ligase and antimicrobial proteins were analyzed. Additionally, for the comparative analysis, the already known inhibitors i.e., tazobactam [35], are also considered for molecular docking. The X-ray crystallographic structure of β-lactamase from *Escherichia coli* str. Sakai (PDB ID: 6N9K) and CTX-M-15 (PDB ID: 7BDS) with a resolution of 1.60 and 0.91 Å were downloaded from Protein Data Bank (PDB) (http://www.rcsb.org/pdb/home/home.do (accessed on 18 November 2022)) [35,36]. The crystallized structure was protonated (adding charges), and energy was minimized (Amber99 force field) using Molecular Operating Environment (MOE) designed by Chemical Computing Group, Montreal, QC, Canada [37,38]. Furthermore, the 3D structures of chemical compounds (**4a**–**h**) and tazobactam were optimized through energy minimization using MMFF94x force field. Target protein structures (6N9K and 7BDS) were docked using placement methods i.e., Triangular Matcher and Induced fit with a scoring method of London dG and GBVI/WSA dG. The Triangular Matcher is a MOE docking placement method and was implemented with three different scoring functions, i.e., London dG, GBVI/WSA, and Affinity dG [39]. It is the best docking algorithm for standard and well-defined binding sites. Moreover, MOE assigns S-score (docking score) based on hydrogen bonds, hydrophobic interactions, salt bridges, solvent exposure, sulfur-LP, and cation-π interaction. The lower the S-score more stabilized the conformation [40,41]. Herein, the algorithm gives the five best poses out of the 30 stochastic search confirmations. The pose with minimum S-score representing the stable energy conformation was selected for further protein-ligand interaction and binding cavity analysis in Discovery Studio version v19.1.0.18287 (BIOVIA, San Diego, CA, USA) [42].

### 3.3. Molecular Dynamic (MD) Simulations

The molecular dynamics (MD) simulations were performed using Gromacs software version 2018 [43]. The docked complexes of two highly active ligands, **4a** and **4c**, with 7BDS and 6N9K, were energy minimized. The ligand topology of **4a** and **4c** was generated using the cgenff server. The four complexes (6N9K-**4a**, 6N9K-**4c**, 7BDS-**4a**, 7BDS-**4c**) were solvated with TIP3P water molecules in a cubic box, and counterions were added to neutralize the system. The systems were then energy minimized using the steepest descent and conjugate gradient algorithms to remove steric clashes and close contacts. The equilibration of the systems was carried out in the NPT ensemble for 100 ps, during which the temperature and pressure were maintained at 300 K and 1 bar, respectively. The production MD simulations were carried out for 100 ns using the CHARMM36 force field with a time step of 2 fs [44]. The trajectories generated from the MD simulations were analyzed to obtain Root-Mean Square deviations (RMSD) of protein-ligand complexes to investigate their conformational stability. All simulations were performed on high-performance computing clusters using parallel computing to improve efficiency.

## 4. Conclusions 

All the *N*-(4-methylpyridin-2-yl)thiophene-2-carboxamides (**4a**–**h**) are good ESBL-producing antibacterial agents, but the compounds **4a** and **4c** show the highest activity. Moreover, compound **4a** followed by **4c** is the most potent inhibitor against the selected target enzymes (6N9K and 7BDS) of β-lactamase *E. coli* in comparison to the remaining *N*-(4-methylpyridin-2-yl)thiophene-2-carboxamide analogues. The reported binding residues were crucial in stabilizing the complexes through hydrogen bonding and hydrophobic (π-interactions) interactions. In conclusion, in vitro and computational studies provide promising evidence for **4a** and **4c** compounds that could be used as novel β-lactamase inhibitors to combat antibiotic resistance.

## Data Availability

Data are contained within the article and Appendix A.

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
