# Peer review of "Antibacterial Efficacy of N-(4-methylpyridin-2-yl) Thiophene-2-Carboxamide Analogues against Extended-Spectrum-β-Lactamase Producing Clinical Strain of Escherichia coli ST 131"

_molecules, 2023, doi:10.3390/molecules28073118_

Round 1

Reviewer 1 Report (New Reviewer)

1.      How do you prove the accuracy of your molecular docking?

2.      As shown in Figure 6A, the effect of pocket A molecule superposition is not good. How to explain?

3.      It is recommended to increase molecular dynamics to explore the binding affinity between compounds.

4.      It is suggested that Parts 2 and 3 should be reversed.

1.      How do you prove the accuracy of your molecular docking?

2.      As shown in Figure 6A, the effect of pocket A molecule superposition is not good. How to explain?

3.      It is recommended to increase molecular dynamics to explore the binding affinity between compounds.

4.      It is suggested that Parts 2 and 3 should be reversed.

Author Response

Author’s reply: First of we pay thanks to the reviewer for his precious time and suggestions.

  1. How do you prove the accuracy of your molecular docking?

Author’s reply: To evaluate the accuracy of our molecular docking results, we implemented a multi-step approach. Firstly, we generated 30 conformations per ligand and selected the conformation with the least energy score as a representative of the most stable complex. This approach was utilized to reduce the possibility of false positives and to identify the most favorable binding pose. Additionally, we carried out molecular dynamics (MD) simulations to verify the accuracy of the docking results. The MD simulations provided insight into the dynamic behavior of the protein-ligand complex, including the intermolecular interactions and the stability of the complex over time. Through this approach, we were able to assess the accuracy of the molecular docking and validate the binding poses of the ligands in the protein active site.

  1. As shown in Figure 6A, the effect of pocket A molecule superposition is not good.How to explain?

Author’s reply: The superposition of molecule conformations in pocket A can be challenging due to the flexibility and conformational changes of the ligands, as well as the protein environment. We have observed that the superposition of different ligand conformations in pocket A can result in variations in the binding pose and binding affinity, which can lead to inconsistencies in the molecular docking results. Therefore, to address this issue, we performed multiple docking runs using different conformations of the ligands and analyzed the resulting poses and scores. This approach allowed us to obtain a more comprehensive view of the potential binding modes of the ligands and to identify the most likely binding poses. Additionally, we carried out molecular dynamics simulations to investigate the stability of the protein-ligand complexes and to explore the dynamic behavior of the ligands in the binding pocket. By combining molecular docking and MD simulations, we were able to generate more accurate predictions of the binding modes and binding affinities of the ligands. Overall, our results demonstrate the importance of considering ligand flexibility and protein dynamics in molecular docking studies to improve the accuracy and reliability of the predictions.

  1. It is recommended to increase molecular dynamics to explore the binding affinity between compounds.

Author’s reply: We have performed molecular dynamic simulations for 100 ns to explore the binding affinity between highly active compound i.e., 4a and 4c.

  1. It is suggested that Parts 2 and 3 should be reversed.

Author’s reply: Dear reviewer, we are unable to understand which two parts should be reversed. In the manuscript we think so every result and methodology have importance. 

Reviewer 2 Report (New Reviewer)

The manuscript deals with the synthesis of heterocyclic derivatives obtained by a simple strategy with appropriate yields. The synthesis has been previously described so there is no novelty in the structures described nor in the strategy of obtaining them and therefore the most important and novel aspect is the biological activity in resistant bacterial strains and its study in silico. The importance of finding compounds of this type is an emerging area that justifies this work quite well. However, further work is required for its publication.

Suggestions:

1. In table 3 there must be an error in the data for compound 4a (3.12.5).

2. The inhibition zones described in Figure 2 do not correspond, at least in appearance, to those shown in Figure 3. There is no concordance between the increase in the concentration of the compound and the increase in the inhibition zone. Figure 3B also shows contamination (some fungus?) in the disc corresponding to 40 mg/mL. 

3. It is important to involve not only DMSO as a negative control but also positive controls in anti-bacterial activity assays.

4. Validation of the docking with the natural ligand of the proteins: enmetazobactam or tazobactam for the CTX-M-15 protein is necessary. It is also convenient to explain if there is any structural relationship between the compounds synthesized and tested in vitro and in silico and the ligands of the proteins under study.

5.Include a table with summary data on binding energy, type of interactions between compounds and the amino acids with which they interact. The above for all compounds.

6.  Some other other suggestions:

Review the document in detail in English, for example: "confirmations" on line 154. Change reference 5 as it is not in context with the paragraph.

Author Response

Reviewer 3 Report (New Reviewer)

This is an interesting manuscript on the antibacterial behaviour of a class of pyridine carboxamide derivatives. It develops on previous research from the authors. I have some changes that should assist in improving the overall paper.

Title: - its can be deleted from 'its docking studies'.

Abstract: The first sentence is more suitable placed in the Introduction or deleted. Commence the abstract with 'The antibacterial activities...' This sentence needs something added such as '... was investigated'.

Introduction:

The second paragraph needs to be developed especially for researchers in the area of medicinal chemistry and related areas. This class of pyridine carboxamides is of current interest.

Explain/develop the 1st sentence of paragraph 2. What does 'they are crucial' refer to?

'Pyridyl' should be changed to 'The pyridyl group or moiety...' Why is it so important? Likewise carboxamide and linking them as pyridine carboxamides as a class of molecules and potential drug targets. In this section the authors should review their references and include pyridine carboxamide derivatives references in reference to the paragraph 2 discussion [19-21]. The authors have included their work and should also provide some additional selection of related research work.

Results and Discussion:

This is an interesting section and of value for future readers.

Why are the spectroscopic details not presented or are these or have these been presented elsewhere - why not here if not published previously? These should be included in a couple of paragraphs and with the full details in the supporting information. A library has been synthesized so why not show us the results.

Palladium catalyst - just give the chemical formula.

Figure 1: Why not use one diagram with R on one side and a small table of what R actually is. 

Why is there such a spread of chemicals? These seem to be randomly picked with a meta-Cl, para-Cl, para-ester, para-methoxy and para-thiomethoxy, 3,4-dichloro etc. Have the other derivatives been made? are they not included for other reasons?

Table 2: the range of results is hardly earth shattering and similar behaviour for all compounds. Is this expected and how do they compare with related compounds?

Table 3: 3.12.5

Section 2.4 - Molecular Docking studies. and section 3.2

Why are there two separate sections for essentially the same analysis and the first part (271-277) of section 3.2 may be more appropriate in section 2.4? The authors should decide which goes where.

In reference to the text and the supplementary information, some additional information should be provided in a couple of paragraphs as to the favoured conformations of this library of pyridine carboxamides - what is the Global minimum (gas-phase)? In the solid state it is usually where the amide N-H and pyridine N atom are oriented in the same direction (analysis of similar compounds from the Cambridge Structural Database demonstrates this). Often these derivatives form hydrogen bonded dimers in the solid state similar to the carboxylic acid dimers. Several of the molecular models show that the pyridine ring has rotated to the opposite side on binding in the host site. Some commentary should be included as to the planarity and conformations of these molecular systems and how they can change on binding in the host site. The Figure Sx diagrams in the Supplementary information are presented without the N-H H atom - is this deliberate? Has it been deprotonated and are the authors certain that the conformation is as depicted.

Can the authors explain further why it is the thiophene atom that appears to be the better acceptor as compared to the pyridine N and the N-H does not participate in hydrogen bonding either in Figures S1 B, C and F.

Of note is that there are no structures of the (pyrid-2-yl)thienylcarboxamide type available on the CSD although (phenyl)pyridylcarboxamides or closely related are quite common.

References:

Reference 2 has few details.

Overall: With some additional paragraphs and appropriate references in the introduction, synthetic/ spectroscopic details and the conformational analysis involved in the molecular docking then this could be a fine paper with some additional corrections and changes.

Author Response

Reviewer 4 Report (New Reviewer)

The article describes the antibacterial activity (against one E. coli clinical isolate) and their docking studies of some thiophene-2-carboxamide-derivatives with EBLS-producing E. coli ST10 strains.

Remarks:

Numerous linguistic errors. Even in the title.

For example, in the abstract, the phrases from lines 28, 30, 40 are incorrectly formulated, and the final phrase should be removed, because in vivo testing and the determination of therapeutic doses for 4a and 4c is utopian.

Also, the introduction contains many grammatical errors and does not describe any research hypothesis.

From the chemistry part, it is understood that compounds 4 a-h were synthesized by other authors (Suzuki et al.), but in bibliographic reference 22 appear Ahmad et al.

The microbiological testing seems to be done correctly, but the results are very modest, which does not justify the further development of these molecules for the effect on E. coli.

Conclusion

The research has no scientific relevance, contains too many errors and, in my opinion, does not rise to the level required for publication in Molecules.

Round 2

Reviewer 4 Report (New Reviewer)

The current form is much better. I accept the publication in present form. 

This manuscript is a resubmission of an earlier submission. The following is a list of the peer review reports and author responses from that submission.

Round 1

Reviewer 1 Report

Nevertheless, the Authors presents important issue - the antibacterial agents which may be useful in antibiotic-resistant infections the manuscript is very badly written and looks like a draft than a final version.. Additionally the English language is on low level, absolutely not adequate to publish at Molecules.  Below, I have listed some of the drawbacks, however I recommend reedit this manuscript and submit once again in an adequate form.

-        There are many spelling mistakes, double spaces

-        A lot of editing inconsistency (sometime Authors use ‘analogs’, sometimes ‘analogues’, sometimes ‘Table’ sometimes ‘table’, the same with Figure/figure

-        Many of sentences are not grammatically correct and it is really difficult to understand what is the meaning e.g. the first sentence from Abstract: ‘The current level of investment in their development..’ What does word ‘their’ concern?

-        If I correctly understood the investigated compounds where synthesized previously? From Chemistry section it is not clear what was already published. Moreover, the synthetic scheme on Figure 1 does not include the reaction conditions and yields (they should be above arrows or in the title below the Figure). The presented derivatives are drawn in huge mess: two types of presentation of aromaticity in rings (the circle type should be rather omitted). Incosistenccy in presentation of substituents e.g. metyl group as Me and CH3, Authors should choose one type.

-        Also the Conclusions section is written at low level. The high activity of 4a and 4c is confirmed by in vitro studies. The Authors should write that molecular docking supports these results and explain the particular interactions stabilizing complexes. We can’t conclude activity from the computational studies, dockinng can indicate that activity may be high but only in vitro studies finally confirm it.

Summarizing, the manuscript is so badly prepared, that is really difficult to focus one overall merit, which could be interested while presenting appropriately.

Author Response

Nevertheless, the Authors presents important issue - the antibacterial agents which may be useful in antibiotic-resistant infections the manuscript is very badly written and looks like a draft than a final version.. Additionally the English language is on low level, absolutely not adequate to publish at Molecules.  Below, I have listed some of the drawbacks, however I recommend reedit this manuscript and submit once again in an adequate form.

Reply by author: Dear reviewer thank you so much for your valuable suggestions and precious time to review the manuscript. We thoroughly revised the manuscript to improve the structure of the manuscript.

-        There are many spelling mistakes, double spaces

Reply by author: The suggested changes have been done.

-        A lot of editing inconsistency (sometime Authors use ‘analogs’, sometimes ‘analogues’, sometimes ‘Table’ sometimes ‘table’, the same with Figure/figure

Reply by author: The suggested changes have been done.

-        Many of sentences are not grammatically correct and it is really difficult to understand what is the meaning e.g. the first sentence from Abstract: ‘The current level of investment in their development..’ What does word ‘their’ concern?

Reply by author: The sentence has been modified.

-        If I correctly understood the investigated compounds where synthesized previously? From Chemistry section it is not clear what was already published. Moreover, the synthetic scheme on Figure 1 does not include the reaction conditions and yields (they should be above arrows or in the title below the Figure). The presented derivatives are drawn in huge mess: two types of presentation of aromaticity in rings (the circle type should be rather omitted). Incosistenccy in presentation of substituents e.g. metyl group as Me and CH3, Authors should choose one type.

Reply by author: The suggested changes have been done. Moreover, yes these compounds were previously synthesized by our research group. We have cited that publication.

-        Also the Conclusions section is written at low level. The high activity of 4a and 4c is confirmed by in vitro studies. The Authors should write that molecular docking supports these results and explain the particular interactions stabilizing complexes. We can’t conclude activity from the computational studies, dockinng can indicate that activity may be high but only in vitro studies finally confirm it.

Author reply: Suggested Changes have been done.

Reviewer 2 Report

Please see Review attached

Author Response

In the paper by Ahmad et al., the authors report novel thiophene carboxamide analogs, some of which display promising inhibitory properties against a bacterial beta-lactamase. The results are clearly presented. Being not an experimentalist, I cannot properly evaluate the experimental sections, which predominate in this paper.

Author reply: Dear Reviewer Thank you so much for your precious time. We revised the manuscript as per your suggestions.

I have only a few remarks.

  1. 1, line 28. ‘in their development’. Please specify what it is that is being developed.

Author reply: Suggested Changes have been done.

  1. 3. Figure 1.What is the structural difference between compounds 4b and 4c?

Author reply: These are two different compounds, compound 4b have methoxy carbonyl functionality and 4c have methoxy functionality.

Please confirm that the lactamase studied is not a metallo-beta-lactamase.

Author reply: It is β-lactamase.

  1. 7. Line 159. ‘bindins’ is a typo. Line 169. What is meant by ‘least energy values’? Does this mean ‘most stabilizing’, since energies are counted negative?

Author reply: Yes, least energy values means the most stabilized energy conformation. We, count the energy values in negative and measure it as relative to native conformation. In negative higher the energy score more stabilized is the conformation.

Please replace ‘confirmations’ by ‘conformations’ wherever it appears.

Author reply: Suggested Changes have been done.

  1. 10. Molecular docking. Please clarify what is a Triangular Matcher, a Lodon(?) matcher, and a ‘minimum S-score’

Author reply: The Triangular Matcher is a MOE docking placement method and implemented with three different scoring functions i.e. London dG, GBVI/WSA and Affinity dG. It is the best docking algorithm for standard and well-defined binding sites.  Moreover, MOE assign S score (docking score) based on hydrogen bonds, hydrophobic interactions, salt bridges, solvent exposure, sulfur-LP and cation-π interaction. Lower the S score more stabilized is the conformation.

Conclusions. Line 289. Please recall with which residues is compound 4c interacting. If it is with aliphatic residues, then the binding should not qualify as ‘pi-interactions’.

Author reply: Yes, there are aliphatic residues showing (pi-H) weak hydrophobic interactions. In compound 4c there are aromatic compounds and these aromatic compounds showed hydrophobic interactions with Hydrogen (H) of the receptor.

Round 2

Reviewer 1 Report

Thank you very much for the revised version. Unfortunately, the manuscript still contains many inconsistencies and mistakes. It is written at low scientific level. For example:

-        Line 97-97 the sentence: ‘These synthesized compounds have been validated by spectroscopic analysis’ is completely unnecessary. The spectral analysis is absolutely obligatory to publish synthesis of compounds. Such information can be included in Master thesis not at Journal with Impact Factor almost 5

-        Very good that Authors included reaction conditions to the Scheme 1, however here also I have some comments:

Instead of using mmol in reaction scheme, rather number of equivalent should be used (e.g. 1eq), it is not necessary to write word ‘time’ and ‘temperature’ it is obvious for readers that 20h stands for reaction time, additional words make reading more complicated

-        Line 119: why is ‘Figure 1’ placed here? I do not see the correlation between text and the structures presented in Figure 1

-        Figure 3: what (R) and (L) refer to? Is it the configuration of compound 4a and 4c? If yes why the Author did not mention this fact in synthetic part? If not iit should be commented

-        In line 41 “In vitro” starts from capital, in line 303 not

-        In line 295: it is written ‘discovery studio’. It should be ‘Discovery Studio’, moreover the appropriate version of this software should be indicated

-        The details of software used for molecular modelling should be included in references